# Ceramides Mediate Insulin-Induced Impairments in Cerebral Mitochondrial Bioenergetics in ApoE4 Mice

**DOI:** 10.3390/ijms242316635

**Published:** 2023-11-23

**Authors:** Sheryl T. Carr, Erin R. Saito, Chase M. Walton, Jeremy Y. Saito, Cameron M. Hanegan, Cali E. Warren, Annie M. Trumbull, Benjamin T. Bikman

**Affiliations:** Department of Cell Biology and Physiology, Brigham Young University, Provo, UT 84602, USA; sherylteresa@gmail.com (S.T.C.); ersaito3@gmail.com (E.R.S.); chase.m.walton@gmail.com (C.M.W.); annie.trumbull@gmail.com (A.M.T.)

**Keywords:** insulin resistance, Alzheimer’s disease, ApoE4, dyslipidemia, ceramides, mitochondrial bioenergetics, cerebral cortex

## Abstract

Alzheimer’s disease (AD) is the most common form of neurodegenerative disease worldwide. A large body of work implicates insulin resistance in the development and progression of AD. Moreover, impairment in mitochondrial function, a common symptom of insulin resistance, now represents a fundamental aspect of AD pathobiology. Ceramides are a class of bioactive sphingolipids that have been hypothesized to drive insulin resistance. Here, we describe preliminary work that tests the hypothesis that hyperinsulinemia pathologically alters cerebral mitochondrial function in AD mice via accrual of the ceramides. Homozygous male and female ApoE4 mice, an oft-used model of AD research, were given chronic injections of PBS (control), insulin, myriocin (an inhibitor of ceramide biosynthesis), or insulin and myriocin over four weeks. Cerebral ceramide content was assessed using liquid chromatography–mass spectrometry. Mitochondrial oxygen consumption rates were measured with high-resolution respirometry, and H_2_O_2_ emissions were quantified via biochemical assays on brain tissue from the cerebral cortex. Significant increases in brain ceramides and impairments in brain oxygen consumption were observed in the insulin-treated group. These hyperinsulinemia-induced impairments in mitochondrial function were reversed with the administration of myriocin. Altogether, these data demonstrate a causative role for insulin in promoting brain ceramide accrual and subsequent mitochondrial impairments that may be involved in AD expression and progression.

## 1. Introduction

Hyperinsulinemia, both the driving cause and clinical manifestation of insulin resistance [1], is the most common metabolic disorder worldwide [2,3]. Appropriately, much of the research on insulin resistance and hyperinsulinemia revolves around insulin’s cardiometabolic roles in the development of type 2 diabetes, hypertension [4], and atherosclerosis [5], among others. However, once thought to be an insulin-independent organ [6,7], it is now widely accepted that the brain is also insulin-responsive and, therefore, is likely susceptible to fluctuations in peripheral insulin and insulin sensitivity. 

Insulin is a hormone secreted by pancreatic β cells that dictates energy utilization within the body and regulates cellular glucose uptake. As a brain growth factor, insulin is necessary in development but is also highly involved in other processes in adulthood [8]. The hormone is transported into the brain via saturable transporters in the endothelial cells of the blood–brain barrier. While the majority of glucose uptake in the brain occurs independent of insulin via glucose transporters GLUT1 and GLUT3 expressed in glia and neurons, respectively [9], regions involved in regulating whole-body energy homeostasis, cognition, and other functions (e.g., hypothalamus, hippocampus and cerebral cortex, olfactory bulb, cerebellum) express the insulin-dependent GLUT4 [10,11]. Indeed, insulin signaling has been shown to play essential roles in synapse density, regulating synaptogenesis and synaptic plasticity [12]. 

Alzheimer’s disease (AD) is the most common form of dementia and is characterized by neurodegeneration that progressively impairs cognition and behavior. The disease affects approximately 6.5 million Americans 65 years and older [13], which is projected to increase to 13.8 million by 2060. Interestingly, the global rise in AD parallels trends in insulin resistance and metabolic syndrome, hinting at an important relationship between peripheral insulin and brain function [14]. Indeed, this relationship has been explored by Kuusisto et al., who suggested the relevance of insulin and glucose metabolism in AD and demonstrated that insulin resistance significantly correlates with AD [15]. It is now widely acknowledged that impairments in brain metabolism, insulin sensitivity, and mitochondrial function are core characteristics of the disease. 

In AD, neurodegeneration follows a pattern such that brain regions involved in learning and memory, such as the hippocampus and entorhinal cortex, display neurodegeneration in early disease stages. However, as the disease progresses, neurodegeneration spreads to other cognitive areas of the brain, including areas of the cerebral cortex responsible for language, decision-making, sociality, and other more basic behaviors. Specifically, the connection between the hippocampus and prefrontal cortex is associated with cognitive dysfunction in AD and represents the link between cognition and emotion, which are both disrupted in AD progression [16]. 

Ceramides are a bioactive family of sphingolipids with structural and functional roles within the cell. Under normal conditions, ceramides found in cell membranes are typically associated with lipid rafts that provide structural support and also have roles in cell signaling that mediate cell growth, proliferation, senescence, and apoptosis, among other processes [17]. However, under obesogenic conditions, ceramides can take on maladaptive roles. Ceramides have been hypothesized to link peripheral adiposity and central insulin resistance, which have substantial implications for AD etiology and treatment [18]. 

Previous studies have demonstrated significant elevations in brain ceramide content in patients with AD and other neurodegenerative disorders [19] and have, as mediators of apoptosis, been suggested to drive neurodegeneration [20,21]. One study demonstrated that a high-fat diet (HFD) increases pro-ceramide gene expression in the liver but not the brain of wildtype C57BL/6 mice [22]. The increase in hepatic ceramide synthesis machinery increased oxidative stress and markers of neurodegeneration in the temporal lobe, suggesting that brain insulin resistance may be mediated by the hepatic production of ceramides that cross the blood–brain barrier and promote apoptosis. However, the effect of the HFD on the accumulation of ceramides in the brain was not directly assessed. 

The accumulation of long-chain ceramides has been observed in post-mortem AD brains [23], and in light of previous findings noted above, we speculated that brain ceramide accrual may be a molecular mechanism of insulin resistance within the brain. In this study, we explore whether hyperinsulinemia is sufficient to increase peripheral adiposity and increase ceramide accumulation within the brain of ApoE4 mice, a model of sporadic AD. We hypothesize that changes in brain ceramide content impair mitochondrial function. The data presented here suggest ceramides may play a causal role in insulin-induced mitochondrial impairment in AD and represent a potential target for future research. 

## 2. Results

### 2.1. Chronic Insulin Injections Increase Body Weight and Reduce Insulin Tolerance in Male and Female ApoE4 Mice 

Over the four weeks of conditioning, insulin injections elicited a significant increase in body weight compared with all other groups at weeks three and four. Myriocin treatment reversed this effect (INS + MYR) (Figure 1a). The chronic insulin injections alone, consistent with our previous work in WT mice [24,25], elicited an elevation in fasting glucose that reverted to control levels with myriocin treatment (Figure 1b). Chronic insulin injections decreased insulin sensitivity, evidenced by a blunted reduction in plasma glucose concentrations following insulin injection in the ITT that was inhibited with MYR treatment (Figure 1c). 

### 2.2. Insulin Increases Brain Ceramide Accrual in Male and Female ApoE4 Mice

To identify changes in brain ceramide concentrations following the induction of hyperinsulinemia, a broad spectrum of ceramide species was assessed. Ceramides with side chains C16:1, C20, C24, and C24:1 were significantly increased with insulin treatment and reverted to control levels when combined with MYR (Figure 2). Together, the increase in C16:1, C20, C24, and C24:1 ceramides yielded a robust increase in total ceramide levels with insulin injections that was rescued with myriocin administration (INS + MYR). Myriocin treatment alone also significantly reduced total ceramide content compared to control PBS-treated mice.

### 2.3. Insulin Disrupts Mitochondrial Function in Brain Tissue of Male and Female ApoE4 Mice

Mitochondrial oxygen consumption rates were measured to assess the effects of hyperinsulinemia on mitochondrial function in our APOE4 mice (Figure 3a). Insulin treatment significantly reduced respiration rates associated with oxidative phosphorylation (i.e., with the addition of ADP; GMP) and complex II-supported respiration (i.e., with the addition of succinate, GMSP) compared to PBS controls. A significant reduction was also observed in respiratory rate, an indicator of general mitochondrial fitness (Figure 3b), and complex II factor, with insulin injections (Figure 3c). Myriocin treatment reverted RCR to control levels but not the CII factor.

### 2.4. Chronic Insulin Injections Direct Brain O_2_ Use towards H_2_O_2_ Production in Male and Female ApoE4 Mice

Absolute rates of H_2_O_2_ production did not vary across the treatment groups (Figure 4a). However, when compared with the amount of O_2_ consumed (Figure 4b), the INS group revealed a roughly 54% increase in H_2_O_2_ production per unit of O_2_ consumed. This significant enhancement in H_2_O_2_ production per unit O_2_ consumption was once again rescued by inhibiting ceramide synthesis with MYR.

## 3. Discussion

The global rise in insulin resistance and metabolic dysfunction poses a substantial threat to cognitive health and increases the risk of developing dementias such as AD [26]. AD is a multifactorial disease involving the interplay of both genetics and the environment [27]. In this study, we explored the intersection of an ApoE4 genetic background and hyperinsulinemia in mice. ApoE4, a variant of the lipoprotein ApoE, is the strongest genetic risk factor for developing AD [28], suggesting a link between disrupted lipid homeostasis and AD. Here, we add support to this link by implicating ceramides, a sphingolipid, in AD mitochondrial dysfunction. 

Although hyperinsulinemia and obesity alone are insufficient to cause AD [18], the data presented in this study demonstrate that hyperinsulinemia is sufficient to increase the accumulation of ceramides within the cortex of ApoE4 mice and significantly impair mitochondrial bioenergetics. These mitochondrial effects were reversed with the systemic administration of myriocin, a potent serine–palmitoyltransferase (SPT) inhibitor, which inhibits ceramide synthesis and demonstrates that hyperinsulinemia-induced mitochondrial impairments were mediated by ceramides. These results agree with previous reports that inhibiting ceramide synthesis improves virtually all metabolic disorders in rodents [29]. Although we did not assess the longitudinal effects of hyperinsulinemia on ceramide accrual and downstream mitochondrial dysfunction, these data demonstrate insulin resistance impairs mitochondrial function through ceramide accrual and suggest both insulin resistance and sphingolipids are relevant to AD. 

We acknowledge that including only ApoE4 and not wildtype mice in this study limited our ability to draw conclusions about the detriment of insulin and the efficacy of myriocin. However, our observation that hyperinsulinemia elevated cortical ceramide content agreed with AD pathology, as previous studies have demonstrated significant elevations in brain ceramides in patients with AD and other neurodegenerative disorders [19]. These studies conclude that the normally tightly controlled regulation of ceramide synthesis is likely lost following neurodegeneration. They suggest that excessive ceramide accrual in the brain of AD patients is a consequence of disease mechanisms. Here, we report that hyperinsulinemia induced elevations in cortex ceramide concentrations that impaired mitochondrial oxygen consumption (Figure 3a) and increased the rate of H_2_O_2_ production (Figure 4) in ApoE4 mice. These effects were reversed by inhibiting ceramide synthesis with myriocin, which demonstrates a causal role for ceramides and insulin in AD mitochondrial dysfunction and suggests that ceramide accrual may be more of an active driver than a mere consequence of the disease. 

Chronic insulin exposure reduced peripheral insulin tolerance (Figure 1c), which supports previous findings that insulin alone, from endogenous or exogenous sources, promotes insulin resistance [24,25,30,31,32]. Myriocin treatment protected against deleterious changes in insulin sensitivity and other indicators of insulin resistance that were assessed.

In the current study, hyperinsulinemia increased cerebral cortex concentrations of C16:1, C20, C24, and C24:1 ceramide (Figure 2). We determined these elevations were due to an increase in de novo ceramide synthesis, as the insulin-induced increase in ceramides was reversed with myriocin treatment, an inhibitor of SPT, the rate-limiting step of de novo ceramide synthesis. From the data presented here, determining whether these ceramides were the result of an increase in central or peripheral de novo synthesis was not possible. However, the significant increase in body mass with insulin treatment (Figure 1a) was due to an increase in peripheral adipose mass. Therefore, the increase in brain ceramides was likely a result of ceramide synthesis in peripheral adipose stores and organ-specific triglyceride pools. This would further support the link between dysregulated peripheral lipid metabolism and Alzheimer’s disease but would require more explicit exploration in the future. 

High plasma ceramide concentrations have been associated with hippocampal atrophy and cognitive impairment in Alzheimer’s disease [33,34,35]. More recent work has demonstrated that the plasma ratio of very long (C22-24) to long (C16-18) chain ceramides has more predictive relevance to Alzheimer’s disease than total ceramide concentrations [36]. They suggest that a lower plasma ratio of very long-chain to long-chain ceramides is associated with a higher risk of AD and may be a means of noninvasively assessing disease risk. In this study, we measured the concentrations of specific ceramide species in the cerebral cortex but not plasma. Because we observed significant increases in total ceramide content in the cortex, it is likely that total ceramide content in the plasma also increased. It is possible that there was a lower ratio of very long-chain to long-chain ceramides in the plasma. However, this would require further study. 

Our findings of compromised mitochondrial respiration (Figure 3) in the brain when ceramides are elevated corroborates earlier work from our lab [37]. Specifically, we previously found that ceramides blunt mitochondrial respiration while increasing the H_2_O_2_ production rate via forced and sustained mitochondrial fission. The prevention of normal mitochondrial dynamics (i.e., frequent fission and fusion) leads to these deleterious and pathogenic changes. 

Together, these data add to the growing body of evidence suggesting AD is a metabolic disorder and can be characterized by impairments in brain energy homeostasis and mitochondrial function. More specifically, this work joins other research that suggests ceramides as drivers of AD and potential markers and drug targets of the disease [38,39,40]. While we did not study the onset of clinically relevant (i.e., neurocognitive and other behavioral) disease symptoms, we demonstrate a causative role for insulin in AD mitochondrial dysfunction via ceramides. We show that hyperinsulinemia is sufficient to increase cortical ceramide content, which has deleterious effects on mitochondrial bioenergetics, and that inhibiting ceramide synthesis is sufficient to reverse these insulin-induced changes. We interpreted these mitochondrial effects as detrimental due to the increase in H_2_O_2_ production rate and ceramide content, which are more closely associated with pathology than protection in the context of Alzheimer’s disease [19]. This work provides an interesting molecular perspective on previously published evidence. Castellano et al. [41] found that insulin resistance correlated with reduced brain glucose use in otherwise healthy people. Additionally, Blazquez et al. [42], after reviewing available studies, implicate insulin resistance as a primary contributor to reduced brain glucose use, including the neurological complications, such as Alzheimer’s disease, that result. 

Ceramides are increasingly recognized as pivotal players in the etiology of myriad cardiometabolic disorders. Indeed, this is so widely acknowledged that some advocate the use of ceramides as a plasma marker of heart disease risk [43]. Moreover, anti-ceramide drugs and interventions are increasingly considered potential viable therapies in mitigating the risk of insulin resistance and related comorbidities [44,45,46]. Given this, particularly in light of our current findings, we suggest that ceramide inhibition be considered a valid strategy for future research focused on reducing the risk of Alzheimer’s disease. 

These findings are especially relevant due to the current widespread nature of metabolic dysfunction and excess adiposity. One study assessing US trends in obesity over recent decades demonstrated that over 50% of young adults (ages 18–25) have overweight or obesity [47], which has substantial implications for neurocognitive health. The results presented here suggest that addressing hyperinsulinemia or ceramide synthesis through pharmacological or lifestyle intervention may be effective in alleviating the cognitive burden of obesity and insulin resistance and be protective against AD. 

## 4. Materials and Methods

### 4.1. Animals

Adult male and female ApoE4 C57BL6 mice (average age of 4.5 months) were group-housed, maintained at 22 ± 1 °C, 60–70% humidity, with a 12 h light–dark cycle, and given ad libitum access to food (LabDiet 5001) and water. Mice were randomly assigned to one of four intraperitoneal injection treatments for 4 weeks: 1. vehicle PBS injections (daily); 2. insulin injections (INS; daily; 0.75 mg/kg); 3. myriocin injections in order to inhibit ceramide synthesis (MYR; thrice weekly; 3 mg/kg); and, 4. insulin and myriocin injections (INS + MYR; as indicated). At the conclusion of the study, brains were removed, and cerebral cortex tissue was isolated and processed according to subsequent analysis (see below). 

The following animal studies were conducted in accordance with the principles and procedures outlined in the National Institutes of Health Guide for the Care and Use of Laboratory Animals and were approved by the IACUC (Institutional Animal Care and Use Committee) at Brigham Young University. Additionally, experiments have been reported in compliance with the ARRIVE guidelines 2.0 on reporting animal experiments.

### 4.2. Insulin Tolerance Test

After four weeks of conditioning, mice underwent insulin tolerance tests. Mice were fasted for six hours and then received an intraperitoneal injection of insulin (0.75 unit/kg of body weight). Blood glucose was determined at baseline (week 0) and weekly until sacrifice, as indicated in the figures using the Bayer Contour glucose meter (Whippany, NJ, USA).

### 4.3. Lipid Analysis

Lipids were analyzed as described previously [24] (with work completed by the J. Prince lab at Brigham Young University; multiple reaction monitoring (MRM) profiles were unavailable due to lab shutdown). To isolate lipids, pellets were suspended in ice-cold chloroform–methanol (1:2), incubated for 15 min on ice, and then briefly vortexed. Aqueous and organic phases were separated by the addition of ice-cold water and chloroform. The organic phase was collected in a fresh vial and dried via vacuum centrifugation (Eppendorf Concentrator Plus, Hamburg, Germany). Lipids were then characterized and quantified using shotgun lipidomics on a Thermo ScientificLTQ Orbitrap XL mass spectrometer (Waltham, MA, USA). A 1.74 μM phosphatidylethanolamine, 1 μM C-17 ceramide, 1 μM tripalmitin internal standard cocktail (1 μL) was spiked into each sample for mass calibration and characterization data alignment. 

### 4.4. Mitochondrial Respirometry 

High-resolution O_2_ consumption was determined at 37 °C in permeabilized brain tissue using the Oroboros O2K Oxygraph (Innsbruck, Austria) with MiR05 respiration buffer as described previously [37,48]. Respiration was determined by all or parts of the following substrate-uncoupler inhibitor-titration (SUIT) protocol [49]: electron flow through complex I was supported by glutamate malate (10 and 2 mM, respectively) to determine O_2_ consumption from proton leak (GML). Following stabilization, ADP (2.5 mM) was added to determine oxidative phosphorylation capacity (GMP). Succinate was added (GMSP) for complex III electron flow into the Q-junction. Complex II-supported ETS was then measured by inhibiting complex I with rotenone (Rot; 0.5 M). Last, residual O_2_ consumption was measured by adding antimycin A (2.5 M) to block complex III action, effectively inhibiting electron flow. This value provides a rate of respiration that is used as a baseline. Following the protocol, samples were lysed for protein quantification (BCA protein assay; ThermoFisher, Waltham, MA, USA). Respiration rates are shown relative to total sample protein.

### 4.5. H_2_O_2_ Emissions

H_2_O_2_ generation was measured using an Amplex Red Hydrogen Peroxide/Peroxidase Assay kit (Molecular Probes; A22188, Eugene, OR, USA) as described previously [50]. A reaction mixture containing 50 μM Amplex Red and 0.1 unit/mL HRP in KRPG (Krebs-Ringer phosphate glucose) buffer was prepared (145 mM NaCl, 5.7 mM sodium phosphate, 4.86 mM KCl, 0.54 mM CaCl_2_, 1.22 mM MgSO_4_, and 5.5 mM glucose). The reaction mixture was pre-warmed in a 96-well plate with 100 μL of mixture per well. A 20 μL aliquot of tissue lysate suspended in KRPG buffer (~1.5 × 10^4^ cells) was added to each well. Samples were incubated for 1 h. Fluorescence was measured with a microplate reader (Molecular Devices; San Jose, CA, USA).

### 4.6. Statistics

Values are reported as means ± the standard errors of the means. Statistical analyses were conducted by one-way analysis of variance (ANOVA) accounting for the four treatment groups. Tukey’s test was used for comparisons between groups within each experiment, and differences were considered statistically significant at *p* < 0.05. 

## Figures and Tables

**Figure 1 ijms-24-16635-f001:**
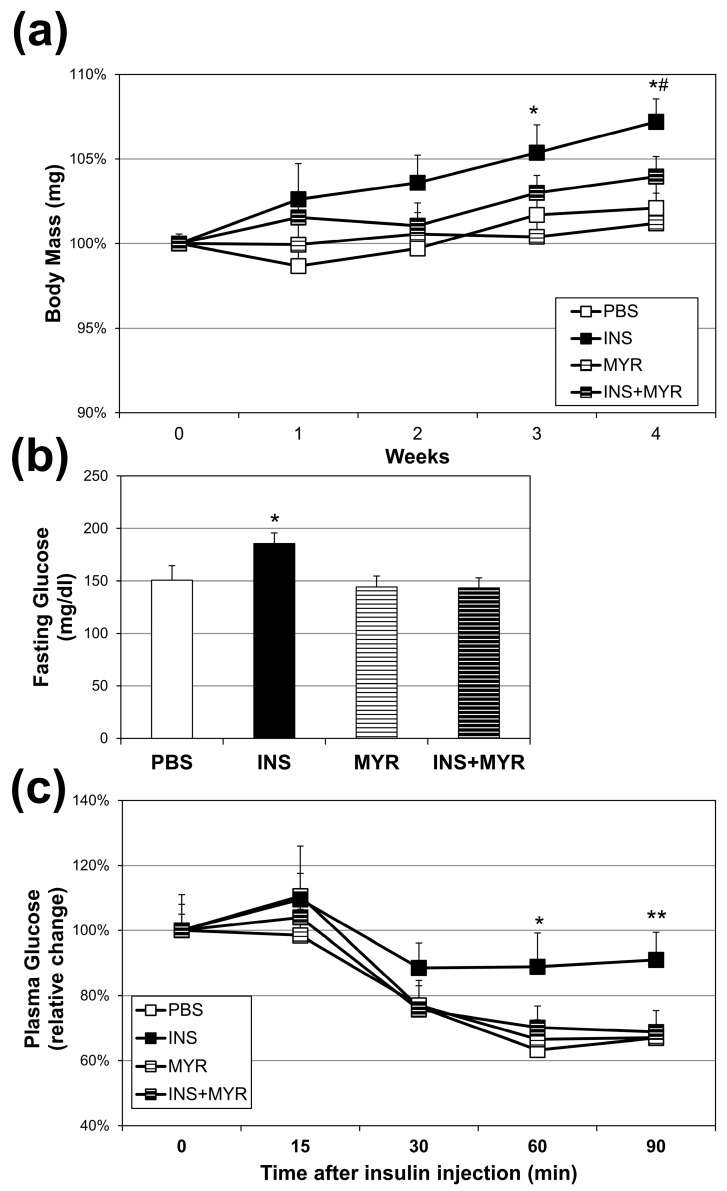
Chronic insulin injections increase body weight and fasting glucose levels and insulin resistance. Male and female ApoE4 mice received injections of PBS (daily), insulin (INS; daily; 0.75 mg/kg), myriocin (MYR, thrice weekly; 3 mg/kg), or INS + MYR. Body weight (**a**) was tracked weekly. Fasting glucose levels (**b**) and insulin tolerance (**c**) were measured after 28 days of treatment. N = 5. * *p* < 0.05 and ** *p* < 0.01 for INS vs. other treatments. # *p* < 0.05 for INS vs. INS + MYR, based on one-way ANOVA with Tukey’s test.

**Figure 2 ijms-24-16635-f002:**
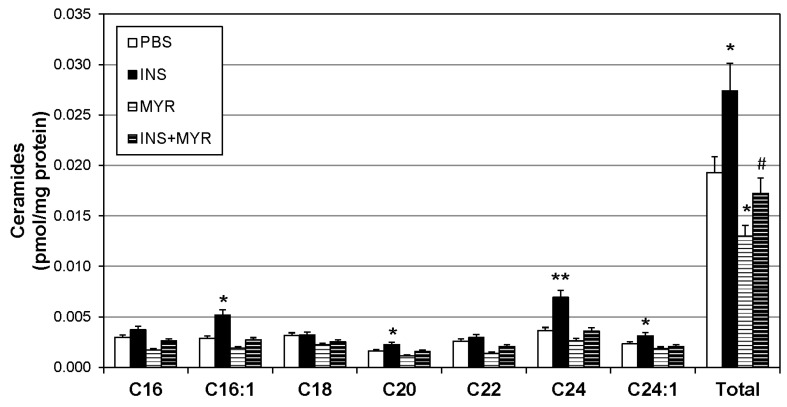
Hyperinsulinemia increases ceramides in the ApoE4 brain. Male and female ApoE4 mice received injections of PBS (daily), insulin (INS; daily; 0.75 mg/kg), myriocin (MYR, thrice weekly; 3 mg/kg), or INS + MYR. Following the 28-day treatment, lipids were isolated from the cerebral cortex for analysis of sphingolipids via liquid chromatography–mass spectrometry. N = 5. * *p* < 0.05 and ** *p* < 0.01 for treatment vs. PBS, # *p* < 0.05 for treatment vs. INS, based on one-way ANOVA with Tukey’s test.

**Figure 3 ijms-24-16635-f003:**
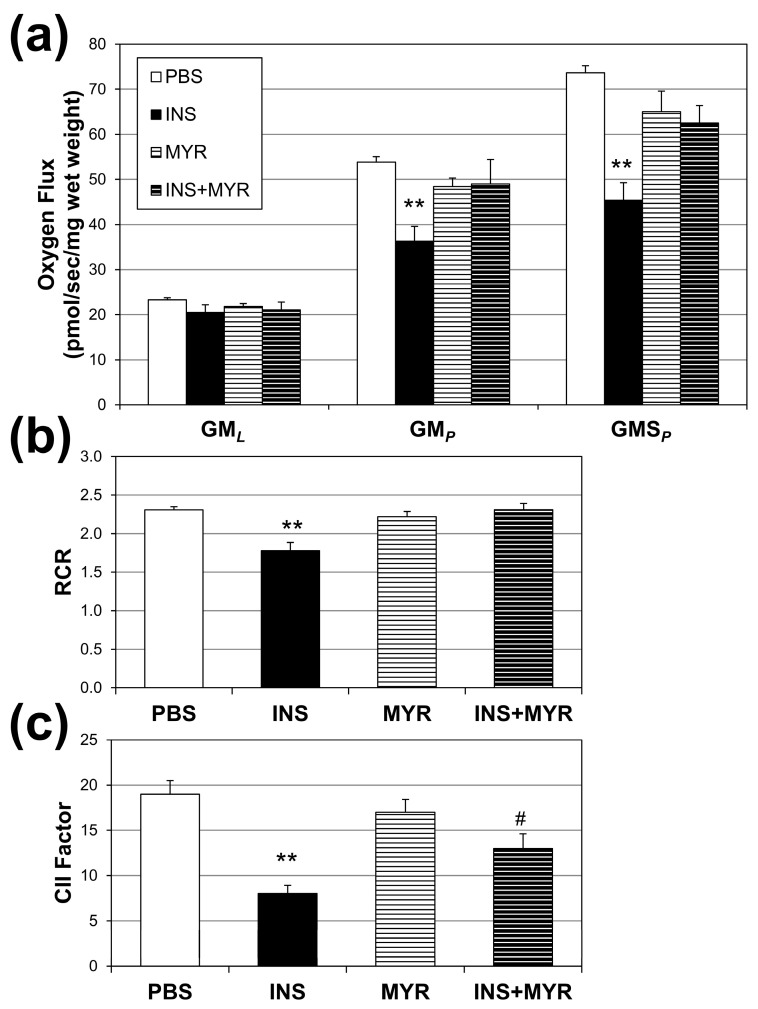
Hyperinsulinemia compromises brain mitochondrial function. Male and female ApoE4 mice received injections of PBS (daily), insulin (INS; daily; 0.75 mg/kg), myriocin (MYR, thrice weekly; 3 mg/kg), or INS + MYR. High-resolution respirometry on cerebral cortex tissue was performed (**a**) using GML (glutamate (10 mM) + malate (2 mM)); GMp: (ADP (2.5 mM)); GMSp (succinate (10 mM)). Respiratory control ratio (RCR; (**b**)) and complex II factor (**c**) were determined, indicated in the Methods. N = 5. ** *p* < 0.01 for treatment vs. PBS, # *p* < 0.05 for treatment vs. INS, based on one-way ANOVA with Tukey’s test.

**Figure 4 ijms-24-16635-f004:**
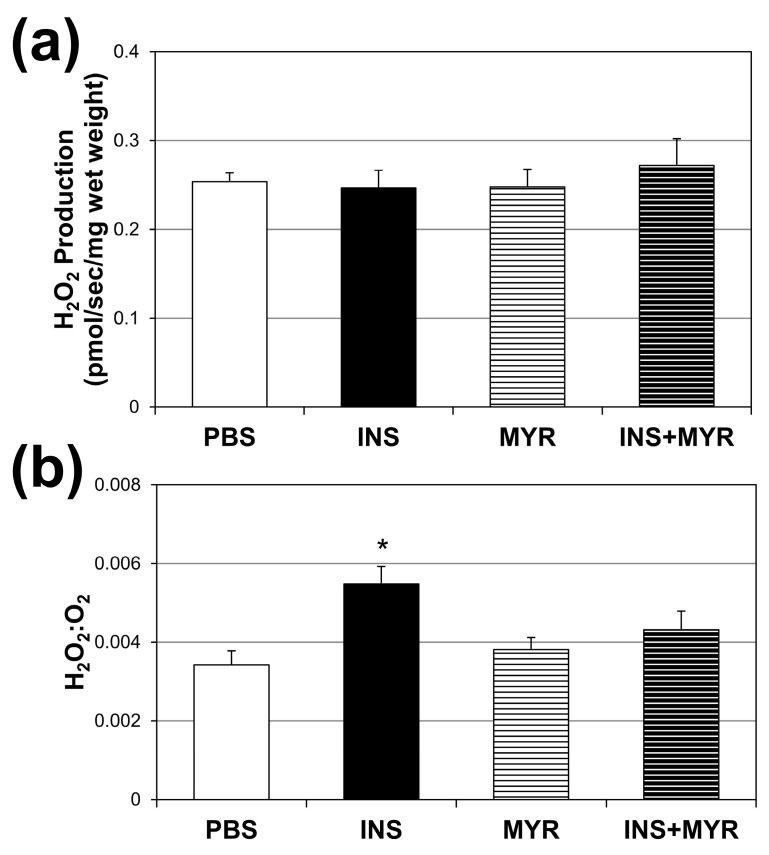
Chronic insulin injections increase brain H_2_O_2_ production. Male and female ApoE4 mice received injections of PBS (daily), insulin (INS; daily; 0.75 mg/kg), myriocin (MYR, thrice weekly; 3 mg/kg), or INS + MYR for 28 days. Cerebral cortex lysate was used to determine fluorescence using Amplex Red (**a**), and the ratio of H_2_O_2_ to O_2_ consumed was determined (**b**). N = 5. * *p* < 0.05 for INS vs. other treatments, based on one-way ANOVA with Tukey’s test.

## Data Availability

The data presented in this study are available on request from the corresponding author. The data are not publicaly available due to data storage method.

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
