# Peer review of "Ceramides Mediate Insulin-Induced Impairments in Cerebral Mitochondrial Bioenergetics in ApoE4 Mice"

_ijms, 2023, doi:10.3390/ijms242316635_

Round 1

Reviewer 1 Report

Comments and Suggestions for Authors

The manuscript entitled: "Ceramides Mediate Insulin-Induced Impairments in Cerebral Mitochondrial Bioenergetics in ApoE4 Mice" by Carr et al. analyzed the effects of insulin and myriocin on mice over four weeks and their impact on cerebral ceramide content, mitochondrial oxygen consumption rates, and H2O2 emissions. The study revealed an increase in ceramide levels in the insulin-treated group, with a reversal of effects observed in the myriocin group.

The paper is well-written and holds significant interest for scientists in the fields of Alzheimer's disease (AD) and neuropathology. It aligns with the scope of the journal. However, several points should be addressed:

Major:

-Please specify the statistical test used in the figure legend.

-How was tissue lysate normalization performed? Was it based on BCA, wet weight, or not normalized at all?

-What was the rationale behind employing ANOVA followed by a student's t-test? Why wasn't a post hoc analysis conducted? It seems that ANOVA results were not reported in the manuscript, so it might be preferable to solely present t-test results. Did the authors check for normal distribution and variance homogeneity before conducting ANOVA/t-tests?

-Include sample size information and report it as n = X beneath each figure legend.

-Specify which internal standard was utilized for the ceramide shotgun lipidomic analysis.

-Add the utilized Multiple Reaction Monitoring (MRMs) into the supplementary material.

-Did the authors account for matrix effects in the mass spectrometry analysis? What were the intra/interday variances?

Minor:

-Correct the spelling error in "4.3 Lipid nalysis" to "4.3 Lipid analysis."

After addressing these points, the manuscript will be ready for publication.

Comments on the Quality of English Language

See minor.

Author Response

  1. We have included statistical methods in the figure legends, including clarifying that Tukey’s test was used to compare ANOVA results.
  2. We have added the following statement in Methods under “4.4 Mitochondrial respiration”:
    1. “Following the protocol, samples were lysed for protein quantification (BCA protein assay; Pierce). Respiration rates are shown relative to total sample protein.”
  3. We were mistaken in the statistical test used. We have clarified this by including mention of Tukey’s test as the posthoc analysis.
  4. Sample size is reported as suggested.
  5. We included the requested information in Methods:
    1. “A 1.74 μM phosphatidylethanolamine, 1 μM C-17 ceramide, 1 μM tripalmitin internal standard cocktail (1 μL) was spiked into each sample for mass calibration and characterization data alignment.”
  6. We do not have any data regarding Multiple Reaction Monitoring. The lipidomics service we utilized did not provide this information and we’ve never received it before on previous services.
  7. We did not account for matrix effects and are unaware of any variances.

Reviewer 2 Report

Comments and Suggestions for Authors

Abstract: 

1. The abstract succinctly captures the essence of the study, but the rationale behind using ApoE4 mice in particular should be more explicitly stated.

2. The term "brain insulin resistance" is introduced but not adequately explained or contextualized for readers unfamiliar with the topic.

Introduction:

3.      The introduction does an effective job of detailing the importance of insulin and its relationship with brain function and AD. However, a clearer transition between insulin resistance, ceramides, and AD could enhance coherence.

4.      The introduction could benefit from more details on why ApoE4 mice were specifically chosen for this study.

5. The discussion successfully ties the findings back to the broader implications in AD research, especially in the context of insulin resistance and ceramide synthesis.

Discussion:

6. Mechanistic Insights: While the discussion successfully ties the findings to broader implications in AD research, it lacks depth in the mechanistic insights behind observed phenomena. For instance, how might increased ceramide accrual mechanistically lead to the observed mitochondrial dysfunctions?

7. Contextualization with Existing Literature: The discussion could benefit from more direct comparisons with existing literature. How do the study's findings align or contrast with previous studies on ceramides, insulin resistance, and AD?

8. Speculative Statements: The section contains statements that might be considered speculative, such as "the increase in brain ceramides was likely a result of ceramide synthesis in peripheral adipose stores." Such statements should either be substantiated with data or appropriately caveated.

9. Future Directions: While limitations are acknowledged, the discussion doesn't delve into potential future research directions that could address these gaps or further elucidate the mechanisms at play.

10. Broader Implications: The study hints at the potential of targeting ceramide synthesis as a therapeutic avenue for AD. However, the discussion could be expanded to address the broader implications of this strategy and the challenges it might pose.

Results-

11. Interpretation of Data: The results briefly state findings, such as increases in brain ceramides or impairments in mitochondrial function, but there is limited interpretation or contextualization of what these findings mean in the broader scope of the study's objectives.

Author Response

  1. We have edited a sentence within the abstract to read (addition in italics): “Homozygous male and female ApoE4 mice, an oft-used model of AD research, were given chronic injections of…”
  2. To account for the additional statement above, we have removed the specific statement of “brain insulin resistance”.
  3. We have included the following statement: “The accumulation of long-chain ceramides has been observed in post-mortem AD brains [23], and in light of previous findings noted above, we speculated that brain ceramide accrual may be a molecular mechanism of insulin resistance within the brain.”
  4. We have included a statement to this effect in the abstract. Due to manuscript length issues (already too long), we’ve not added additional description.
  5. Thank you.
  6. Regarding mechanism, we have included the following section in Discussion:
    1. “Our findings of compromised mitochondrial respiration (Figure 3) in the brain when ceramides are elevated corroborates earlier work from our lab [40]. Specifically, we previously found that ceramides blunt mitochondrial respiration, while increasing H2O2 production rate, via forced and sustained mitochondrial fission. The prevention of normal mitochondrial dynamics (i.e., frequent fission and fusion) leads to these deleterious and pathogenic changes.”
  7. We have included the following statements:
    1. “This work provides an interesting molecular perspective into previously published evidence. Castellano et al. [42] found that insulin resistance correlated with reduced brain glucose use in otherwise healthy people. Additionally, Blazquez et al. [43], after reviewing available studies, implicate insulin resistance as a primary contributor to reduced brain glucose use, including the neurological complications, such as Alzheimer’s disease, that result.”
  8. We have deleted that paragraph.
  9. We included a statement of mechanisms (mentioned in 6 above), but have not detailed any description of future work simply because this is a topic we are actively discussing with the lab (i.e., we don’t have a clear follow-up yet).
  10. We have included the following paragraph:
    1. “Ceramides are increasingly recognized as pivotal players in the etiology of myriad cardiometabolic disorders. Indeed, this is so widely acknowledged that some advocate the use of ceramides as a plasma marker of heart disease risk [44]. Moreover, anti-ceramide drugs and interventions are increasingly considered potential viable therapies in mitigating the risk of insulin resistance and related comorbidities [45-47]. Given this, particularly in light of our current findings, we suggest that ceramide inhibition be considered a valid strategy for future research focused on reducing the risk of Alzheimer’s disease.”
  11. We are unsure of how to address this comment. We believe we’ve provided sufficient context and relevance of the findings in the Discussion.

Round 2

Reviewer 1 Report

Comments and Suggestions for Authors

The authors adressed most of my comments.

However some points are still unclear:

From the authors reply: "

  1. We do not have any data regarding Multiple Reaction Monitoring. The lipidomics service we utilized did not provide this information and we’ve never received it before on previous services.
  2. We did not account for matrix effects and are unaware of any variances."

The authors state that they utilize a lipidomics analysis service. However they don't state which service they utilize. They state which mass spectrometer is used for that but its not clear for the reader if they analyze it themself or use a third party service. Please indicate which service is used. Since using a non deuterized standard and using SHOTGUN lipidomics it is very importanted to see the MRMs and check for isobar lipids. Matrix effects and intrayday/interday variances would be interesting to know. However if the authors really rely on a external service for those measurements the authors need to state where those measurements are done.

After commenting on this issue the manuscript is ready to be published.

Author Response

We engaged (and paid for) a collaborating lab at our university (Dr. John T. Price; Chemistry/Biochemistry) to conduct the lipidomics work. The lipidomics work was completed prior to COVID lockdowns. Regrettably, Dr. Price left the university during this time. I suspect we would be able to obtain these data somehow if he were still here, though we've never been asked for it before this publication. We are unable to fulfill this request. 

Reviewer 2 Report

Comments and Suggestions for Authors

N/A

Author Response

Thank you for the your time and attention on reviewing our manuscript. 

Round 3

Reviewer 1 Report

Comments and Suggestions for Authors

Please add the statement which laboratory has done the lipidomics analysis to the M&M section and add a caveat. After this the manuscript is ready to be published.

Author Response

We have added the following statement: "(with work completed by the J. Prince lab at Brigham Young University; multiple reaction monitoring (MRM) profile unavailable due to lab shutdown)"